# Estimating the impact of drug use on US mortality, 1999-2016

**Dana A. Glei** [1] *, **Samuel H. Preston** [2]

**1** Center for Population and Health, Georgetown University, Washington, DC, United States of America,
**2** Sociology, University of Pennsylvania, Philadelphia, PA, United States of America

* dag77@georgetown.edu

**Data Availability Statement:** All the data used in this analysis are either publicly available (mortality multiple cause-of-death data files for 1981-2004: https://www.cdc.gov/nchs/nvss/mortality_public_use_data.htm; population estimates for 1981-89:

## Abstract

The impact of rising drug use on US mortality may extend beyond deaths coded as drug-related to include excess mortality from other causes affected by drug use. Here, we estimate the full extent of drug-associated mortality. We use annual death rates for 1999–2016 by state, sex, five-year age group, and cause of death to model the relationship between drug-coded mortality—which serves as an indicator of the population-level prevalence of drug use—and mortality from other causes. Among residents aged 15–64 living in the 50 US states, the estimated number of drug-associated deaths in 2016 (141,695) was 2.2 times the number of drug-coded deaths (63,000). Adverse trends since 2010 in midlife mortality are largely attributable to drug-associated mortality. In the absence of drug use, we estimate that the probability of dying between ages 15 and 65 would have continued to decline after 2010 among men (to 15% in 2016) and would have remained at a low level (10%) among women. Our results suggest that an additional 3.9% of men and 1.8% of women died between ages 15 and 65 in 2016 because of drug use. In terms of life expectancy beyond age 15, we estimate that drug use cost men 1.4 years and women 0.7 years, on average. In the hardest-hit state (West Virginia), drug use cost men 3.6 and women 1.9 life years. Recent declines in US life expectancy have been blamed largely on the drug epidemic. Consistent with that inference, our results imply that, in the absence of drug use, life expectancy at age 15 would have increased slightly between 2014 and 2016. Drug-associated mortality in the US is roughly double that implied by drug-coded deaths alone. The drug epidemic is exacting a heavy cost to American lives, not only from overdoses but from a variety of causes.

## Introduction

The drug epidemic has taken a heavy toll in America. Death rates from drug poisoning increased rapidly from 2000 to 2015 in all regions and among all racial and ethnic groups [1]. More than 70,000 Americans died in 2017 as a result of drug overdose [2]. However, the full impact of the drug epidemic on US mortality may extend beyond deaths resulting directly from an overdose. In addition to the obvious connection with accidental poisoning, drug use may increase the risk of dying from other disease and injury processes in ways that are not

https://wonder.cdc.gov/wonder/sci_data/census/inter/type_txt/nchsinte.asp; population estimates for 1990-2016: http://wonder.cdc.gov/bridged-race-v2016.html) or can be obtained from the National Center for Health Statistics by special request (mortality multiple cause-of-death data files with state identifiers for 2005-2016: https://www.cdc.gov/nchs/nvss/dvs_data_release.htm). The restricted-use data cannot be shared publicly because they were provided by a third-party (NCHS). The lead author (DAG) signed the data use agreement with NCHS on May 3, 2018, and obtained the restricted-use data files on May 25, 2018. DAG managed the data and performed all the analyses for this paper.

**Funding:** This work was supported by grant 1R01AG060115 to the University of Pennsylvania from the National Institute on Aging and by the Graduate School of Arts and Sciences, Georgetown University.

**Competing interests:** The authors have declared that no competing interests exist.

recognized in assignments of underlying or even contributing cause of death. A meta-analysis indicated that standardized mortality rates among opioid-dependent individuals are almost 15 times those of the general population; in addition to drug overdose, the most common causes of death among this group were AIDS, trauma, suicide, and liver-related causes (including viral hepatitis), and to a lesser extent, cardiovascular disease, cancer, other digestive diseases, and respiratory diseases [3]. Other studies suggest that drug users show elevated mortality from infectious diseases [4–6], particularly HIV/AIDS [6, 7] and viral hepatitis [6]; respiratory diseases [5, 6]; external causes [4, 6], especially suicide [6]; mental/behavioral disorders [5]; digestive diseases [5, 6]; circulatory disease [5, 6]; and cancer [5, 6].

A Finnish study found that drug-*coded* deaths (i.e., the underlying cause was drug poisoning—regardless of intent—or a drug-related mental/behavioral disorder) accounted for only 36% of all drug-associated mortality, which includes all deaths where drugs were either the underlying or a contributing cause (i.e., the underlying cause was not drug-coded, but one of the contributing causes was a drug-related mental/behavioral disorder) [8]. Among deaths in which drugs were a contributing cause, the most common causes of death were accidents (30%), suicides (30%), and illnesses (31%), primarily diseases of the circulatory, respiratory, and digestive systems [8].

These associations suggest that, in addition to its direct effect on deaths from poisoning, drug use may inflate mortality resulting from infectious diseases, respiratory diseases, external causes, mental/behavioral disorders, digestive diseases, circulatory diseases, and neoplasms. In this study, we use annual death rates by state to model the relationship between drug-coded mortality—which serves as our indicator of the population-level prevalence of drug use—and mortality from other causes. Drug-coded mortality includes deaths from all drugs, medicaments, and biological substances (e.g., opioids, cannabinoids, sedatives/hypnotics, cocaine, other stimulants, hallucinogens, volatile solvents, and other psychoactive substances).

There are various biophysical mechanisms through which drug use might increase risk of mortality from other causes such as circulatory, respiratory, and digestive diseases as well as external causes that are not explicity coded as drug-related. Drug use can have direct effects on the circulatory system (e.g., abnormal heart rate, increased blood pressure, increased risk of heart attack, collapsed veins and bacterial infections of the blood vessels and heart valves from injection drug use) [9, 10]. The American Heart Association calls cocaine the "perfect heart-attack drug" because it increases blood pressure, stiffens arteries, and thickens heart muscle walls, all of which increase the risk of myocardial infarction [10]. Many drugs also affect the respiratory system. Opioids are a central nervous system depressant, reducing the activity of the neurons in the brain and spinal cord, which can inhibit respiratory function by slowing breathing; chronic use of opioids can exacerbate existing respiratory conditions (e.g., emphysema, bronchitis, and asthma) and increase the risk of developing pulmonary edema [11, 12]. Smoking marijuana or crack cocaine can also cause lung damage and severe respiratory problems [12]. Some drugs (e.g., heroin, inhalants, and steroids) can also cause substantial liver damage, especially when combined with alcohol [13]. The effects of drug use on the prefrontal circuits in the brain can impair judgment, thereby increasing risky behavior (e.g., driving under the influence, unprotected sex, needle/syringe sharing) [14] that heightens risk of accidents, injury, trauma, and infectious disease (e.g., HIV/AIDS, viral hepatitis B and C) [15, 16].

A challenge in using individual-level data to estimate the impact of drug use on mortality is that the association between substance use and mortality may result partly from selection: drug users are not randomly selected, so individual-level associations are subject to confounding by unmeasured or poorly measured variables that influence both drug use and mortality. For example, there are high levels of comorbidity between substance use and mental disorders [17, 18]. Drug users may also be more likely to engage in other risky health behaviors [4, 5, 8]

and be less likely to adopt health-promoting behaviors [19] or obtain high-quality health care [20].

There are other reasons why it is difficult to estimate directly the effects of drug use on mortality based on individual-level survey data. First, survey respondents may be unwilling or unable to accurately report/recall substance use. Second, given rapid changes in the availability and dissemination of various substances, it would be difficult to capture use of all relevant substances at baseline and track the dynamics of drug use status [21]. Finally, drug use and drug-related mortality are still relatively rare, thus one would need a very large sample to obtain sufficient statistical power to detect the patterns of interest (see also [22]).

Here, we use aggregate-level data to derive population-level estimates—for the 50 US states—of drug-associated mortality. We examine variation in mortality over time, space, and age to identify the relationship between drug-coded mortality—which serves as an indicator of the population-level prevalence of drug use—and mortality from all other causes of death. In particular, we expect that demographic groups (defined by age, sex, period, and state of residence) that exhibit higher death rates from drug-coded causes of death also experience excess mortality from other causes affected by drug use. We expect this relationship because the frequency of drug-coded deaths is a strong indicator of the prevalence of drug use in a demographic group and drug use is associated with excess mortality from many causes of death.

Our indirect approach does not require identification of individual drug users nor does it rely solely on cause of death coding to determine whether a given death is drug-related. Instead, we use a macro-level statistical model to indirectly estimate the number of "other" drug-associated deaths based on the observed relationship between drug-coded mortality and the aggregate of all other causes of deaths, after controlling for levels and trends in background mortality. This method eliminates recall bias, confounding that results from selection, the challenges of identifying all relevant drugs, and constraints on statistical power resulting from limited sample size since our dataset covers the entire US population. A related approach has been used to estimate smoking-associated mortality based on lung cancer death rates [23–25].

In this paper, we adapt that approach to estimate the full extent of drug-associated mortality in the US. Our results suggest that the impact of drug use on US mortality is substantially greater than drug overdose rates would suggest.

## Methods

### Data

We use annual death counts for each of the 50 US states by sex, five-year age group (to 100+), and cause of death for the period 1999–2016 from the Multiple Cause-of-Death micro-data files compiled by the National Center for Health Statistics (NCHS) [26]. Data for 1999–2004 came from the public-use files; data with state identifiers for the period since 2005 were obtained from NCHS by special request. Corresponding mid-year population estimates by state, sex, and five-year age group (to 85+) also came from NCHS [27]. We restrict our analysis to ages 15 and older because drug-coded mortality rates are very low below age 15 and thus, our model coefficients would likely be unstable at those ages. In order to match the age format of the population estimates, we aggregated deaths at ages 85 and older. Deaths of unknown age ($n = 4,028$ out of 41,156,760 deaths) have been excluded from the analysis. This study is exempt from human subjects review because it involves analysis of existing, deidentified data.

### Modeling strategy

Using negative binomial regression (to allow for possible over-dispersion), we model the relationship between drug-coded mortality rates (classified by age, sex, year of death, and state)

and the corresponding death rates from all other causes. The outcome variable is the number of deaths from all causes other than drugs for a given state-year-age group-sex divided by the number of person-years of exposure (i.e., the death rate, $M_{-D}$). We use negative binomial regression to model the expected value of the logged death rate ($\ln M_{-D}$), fit separately by sex:

$$\ln M_{-D} = \beta_a X_a + \beta_s X_s + \beta_t T + \beta_D M_D + \beta_{Da}(M_D \times X_{a2}). \qquad (1)$$

The model includes a set of control variables designed to capture levels and trends in background mortality (i.e., factors that affect mortality but are unrelated to drug use): $X_a$ represents a set of dummy variables for each age group to account for the age pattern of mortality, which is assumed to be fixed over time and across states; $X_s$ is a set of dummy variables for each state to capture across-state differences in overall mortality levels, which are assumed to be fixed over time; and $T$ is a linear term for calendar year (centered at 1999) to account for the underlying trend in mortality. The key predictor of interest is the drug-coded mortality rate ($M_D$), which is interacted with age by including the product of $M_D$ and $X_{a2}$ (i.e., a set of dummy variables for ages 25–29, 30–34, . . .60–64, and 65+) to allow the relationship between drug-coded mortality and mortality from other causes to vary across age groups. See S1 Appendix for additional details.

Our cause of death assignments are based on ICD-10, which has been in use in the US since 1999. Given the difficulty of distinguishing between unintentional and intentional poisonings [28, 29], we define drug-coded mortality to include deaths in which the underlying cause was coded as drug poisoning—regardless of intent (ICD-10: X40-X44, X60-64, X85, Y10-Y14)—as well as deaths coded to a drug-related mental/behavioral disorder (ICD-10: F11-F16, F18, F19). Our measure of drug-coded deaths includes fatalities from all drugs (i.e., nonopioid analgesics, antipyretics, and antiheumatics; antiepileptic, sedative-hypnotic, antiparkinsonism, and psychotropic drugs; narcotics and psychodysleptics [hallucinogens], including cannabis, cocaine, LSD, mescaline, and opioids such as codeine, heroin, methadone, morphine, and opium; other drugs acting on the autonomic nervous system; and other and unspecified drugs, medicaments, and biological substances).

Our ability to identify the imprint of drug use on mortality rates relies on the fact that drug-coded mortality rates vary across time, state, and age in ways that are not closely associated with the trends, geographic disparities, and age patterns of background mortality. For example, drug-coded mortality increased dramatically over time, whereas overall mortality rates typically declined or changed very slowly. The age pattern of drug mortality, which is highest in midlife, also differs from overall mortality, which increases monotonically with age above midlife. Similarly, states with high drug-related mortality vary in their levels of mortality from other causes (e.g., in 2016, men in Kentucky and Delaware had similarly high levels of drug-coded mortality, but all-cause mortality was much higher in the former than in the latter; men in Mississippi and Wyoming had similarly low levels of drug-coded mortality, but all-cause mortality in Mississippi was even higher than in Kentucky, while all-cause mortality in Wyoming was lower than in Delaware). If levels of drug-coded mortality did not vary over time, age, and place in patterns that differ from those of background mortality (i.e., if drug mortality were highly correlated with background mortality), there would be a substantial risk of spurious correlations when implementing our model. The fact that background mortality has very different features increases our confidence that we are identifying excess mortality that is associated with drug use.

Analyses were performed using Stata 14.2 [30]. We use a robust (i.e., Huber/White/sandwich) variance estimator to compute the standard errors.

**Model elaborations.**   Because of the possible overlap between substance use and smoking and the major effect of smoking on mortality, we explore an alternative model that simultaneously models the association of mortality from other causes with both drug use (for which drug-coded mortality is the proxy) and smoking (for which lung cancer mortality is the proxy). To identify lung cancer deaths, we use the same ICD codes (ICD-10: C33, C34) used in an earlier application[23, 24] (See S2 Appendix for detailed model specification.)

In order to investigate the other causes of death through which drug use influences all-cause mortality, we estimate separate models for ten mutually exclusive and exhaustive cause of death categories. The model specification is identical to that described in S2 Appendix but with mortality from individual causes replacing all-cause mortality as dependent variables. The categories employed are: 1) non-drug-related mental/behavioral disorders; 2) ill-defined causes; 3) infectious/parasitic diseases; 4) respiratory diseases; 5) digestive diseases; 6) non-drug-related external causes; 7) circulatory diseases; 8) neoplasms (except lung cancer); 9) endocrine, nutritional, and metabolic diseases; and 10) all other causes (see S5 Appendix for details). Any inter-state variation in diagnostic or coding practices with respect to whether a death is drug-coded or assigned to another category would contribute to a negative relationship between death rates from drug-coded mortality and mortality from that other cause. For example, if drug overdose deaths are sometimes miscoded to ill-defined causes, it may create a negative association between drug-coded mortality and ill-defined causes. Given high co-morbidity between mental illness and drug use [31], it is also possible that variation in diagnosis and coding practices could create a negative relationship between drug-coded mortality and mortality from non-drug-related mental/behavioral disorders.

We examine the sensitivity of the results to the specification for the time trend in background mortality. In place of the linear time trend, we substitute year fixed-effects (i.e., dummies for each year), allowing for non-linearities in the functional form.

**Estimating the drug-associated fraction.**   For each state-year-sex-age group, we first compute the estimated number of "other" deaths based on the negative binomial regression model given the observed level of drug-coded mortality. At ages 65 and older, the drug coefficient was negative (Table 1), suggesting a substitution effect (i.e., higher levels of drug-coded mortality are associated with lower levels of "other" drug-associated mortality) that could result from interstate variation in reporting (e.g., medical examiners/physicians in some states may be more reluctant than in those in other states to ascribe a death to drugs). Thus, above age 65, the negative number of drug-associated deaths resulting from other causes will offset the number of drug-coded deaths, which are always positive.

We then estimate the predicted number of deaths in the absence of drug use by setting drug-coded mortality to zero. Finally, we divide the difference by the estimate based on the observed level of drug-coded mortality to obtain the fraction of "other" deaths that are drug-associated (see S3 Appendix for details).

**Estimating life expectancy at age 15 in the absence of drug use.**   We use the sex-specific mortality rates by five-year age groups and apply standard life table methods [32] to estimate observed life expectancy at age 15 by sex for each state and for the US as a whole for each year from 1999 to 2016. From the period life tables, we also estimate the observed probability of dying between age 15 and 65.

To estimate these same quantities in the absence of drug use, we multiply the observed mortality rates (by sex, age group, year, and state/US) by one minus the corresponding fraction of all-cause deaths associated with drug use, based on the model that includes smoking. We then recompute the life tables using these adjusted rates. (See S4 Appendix for details.) While this is a standard strategy for estimating the effect of removing a cause of death on life expectancy [32], it assumes that mortality from other causes is unchanged. However, if drug deaths were

**Table 1. Estimated drug coefficients and number and percentage of drug-coded and drug-associated deaths, by sex and age group, US.**

| Sex and Age Group | Estimated Drug Coefficients[a] | | | | Drug-Coded,[b] No. (%) | | Drug-associated (estimated),[c] No. (%) | |
|---|---|---|---|---|---|---|---|---|
| | Model 1 | | Model 2 | | | | | |
| | Coef | 95% CI | Coef | 95% CI | 1999 | 2016 | 1999 | 2016 |
| **Men** | | | | | | | | |
| Ages 15–24 | 0.18 | (0.08, 0.29) | 0.17 | (0.07, 0.28) | 979 (4) | 4,008 (17) | 1,173 (5) | 4,712 (20) |
| Ages 25–29 | 0.32 | (0.25, 0.39) | 0.31 | (0.24, 0.38) | 1,065 (9) | 5,479 (28) | 1,444 (12) | 7,357 (38) |
| Ages 30–34 | 0.35 | (0.29, 0.42) | 0.35 | (0.28, 0.41) | 1,508 (10) | 5,920 (29) | 2,186 (14) | 8,413 (41) |
| Ages 35–39 | 0.29 | (0.23, 0.35) | 0.30 | (0.24, 0.35) | 2,227 (9) | 5,466 (24) | 3,393 (14) | 7,961 (35) |
| Ages 40–44 | 0.24 | (0.18, 0.29) | 0.26 | (0.21, 0.32) | 2,651 (8) | 4,360 (17) | 4,421 (13) | 6,761 (26) |
| Ages 45–49 | 0.25 | (0.20, 0.31) | 0.31 | (0.26, 0.36) | 2,105 (5) | 4,623 (12) | 4,586 (11) | 8,991 (23) |
| Ages 50–54 | 0.39 | (0.35, 0.43) | 0.46 | (0.41, 0.50) | 993 (2) | 4,791 (7) | 3,416 (7) | 15,302 (23) |
| Ages 55–59 | 0.43 | (0.38, 0.48) | 0.57 | (0.52, 0.63) | 437 (1) | 4,338 (4) | 2,507 (4) | 22,090 (22) |
| Ages 60–64 | 0.32 | (0.26, 0.38) | 0.52 | (0.46, 0.59) | 208 (0) | 2,516 (2) | 1,604 (2) | 16,920 (14) |
| Ages 65+ | -0.10 | (-0.16, -0.04) | -0.08 | (-0.14, -0.02) | 425 (0) | 1,765 (0) | N/A | N/A |
| Ages 15–64 | | | | | 12,173 (4) | 41,501 (9) | 24,731 (7) | 98,508 (22) |
| **Women** | | | | | | | | |
| Ages 15–24 | 0.24 | (-0.03, 0.50) | 0.26 | (-0.00, 0.53) | 370 (5) | 1,541 (18) | 411 (5) | 1,685 (20) |
| Ages 25–29 | 0.74 | (0.57, 0.91) | 0.76 | (0.59, 0.93) | 384 (7) | 2,150 (29) | 524 (10) | 2,880 (38) |
| Ages 30–34 | 0.77 | (0.64, 0.90) | 0.79 | (0.65, 0.92) | 615 (8) | 2,477 (25) | 928 (12) | 3,699 (38) |
| Ages 35–39 | 0.59 | (0.48, 0.69) | 0.64 | (0.54, 0.75) | 1,002 (8) | 2,601 (21) | 1,632 (12) | 4,100 (33) |
| Ages 40–44 | 0.45 | (0.36, 0.53) | 0.51 | (0.43, 0.60) | 1,180 (6) | 2,329 (14) | 2,052 (11) | 3,941 (25) |
| Ages 45–49 | 0.47 | (0.41, 0.53) | 0.50 | (0.43, 0.56) | 916 (4) | 2,877 (11) | 1,897 (8) | 5,728 (22) |
| Ages 50–54 | 0.51 | (0.45, 0.58) | 0.53 | (0.47, 0.59) | 565 (2) | 3,165 (7) | 1,532 (5) | 8,373 (20) |
| Ages 55–59 | 0.33 | (0.26, 0.41) | 0.47 | (0.40, 0.53) | 299 (1) | 2,785 (4) | 1,030 (2) | 8,873 (14) |
| Ages 60–64 | -0.08 | (-0.17, 0.02) | 0.21 | (0.11, 0.32) | 154 (0) | 1,574 (2) | 435 (1) | 3,908 (5) |
| Ages 65+ | -0.13 | (-0.20, -0.06) | -0.12 | (-0.19, -0.04) | 550 (0) | 1,481 (0) | N/A | N/A |
| Ages 15–64 | | | | | 5,485 (3) | 21,499 (8) | 10,441 (5) | 43,187 (16) |

N/A Results not shown in cases where the value is negative.

[a] These coefficients are based on sex-specific negative binomial regression models that control for state-level fixed effects, age (categorical), and calendar year (linear); standard errors are computed using the robust (Huber/White/sandwich) variance estimator. Model 1 regresses the death rate from causes other than drugs on the drug-coded mortality rate (see S1 Appendix), whereas Model 2 adds lung cancer mortality as a predictor and the outcome is modified to represent the mortality rate from causes other than drugs or lung cancer (see S2 Appendix). The coefficients shown here correspond to values $\beta'_D$ defined in S3 Appendix. This coefficient implies that a 0.001 change in the drug-coded mortality rate increases the mortality rate for other causes by a factor of $(e^{\beta'_D})$ for the specified sex-age group. For example, $\beta'_D = 0.57$ for men aged 55–59 in Model 2 implies that a 0.001 increase in drug-coded mortality would increase mortality from other causes by 77%. See S1 Table for the full results from the models.

[b] Includes deaths where the underlying cause was coded as either drug poisoning—regardless of intent (ICD-10: X40-X44, X60-64, X85, Y10-Y14)—or drug-related mental/behavioral disorder (ICD-10: F11-16, F19).

[c] Includes drug-coded deaths as well as deaths from other causes (i.e., underlying cause was not drugs or lung cancer) estimated to be drug-associated based on Model 2.

eliminated, it could change the characteristics of the population because individuals dying of drug-associated causes may be a selective group. If those characteristics are correlated with mortality, then death rates from other causes might change, and thus our estimate of the reduction in mortality rates if drug use were eliminated would be biased.

## Results

The dataset on which we fit our models includes 27,000 observations (50 states x two sexes x 15 age groups x 18 calendar years). It represents the universe of 44,152,732 deaths (including

667,196 drug-coded deaths and 2,813,665 lung cancer deaths) occurring to residents aged 15 and older living in the 50 US states during 1999–2016 and comprises 4,333,491,018 person-years of exposure for that population.

Fig 1 shows that drug-coded mortality rates are highest between ages 25 and 59, while rates are low below age 20 and above age 65. In 1999, drug-coded mortality peaked in the early 40s for both sexes, but as rates increased over time, the distribution shifted to younger ages among men, peaking in the 30s in 2016, and to older ages among women, peaking in the early 50s.

Drug-coded mortality in the US generally increased monotonically between 1981 and 2016 (S1 Fig), although acceleration in the rates began earlier for men (1990s) than for women (around 2000). The magnitude of the drug epidemic varies by state. In this same graph, we show the trend for the states with the highest (West Virginia) and lowest (Nebraska) drug-coded mortality rates in 2016. In 1981, drug-coded mortality rates in both of these states were lower than in the US as a whole, but rates increased sharply in West Virginia after 2000. They also grew steadily in Nebraska over the same period, albeit at a much slower pace.

Table 1 shows our estimated coefficients for drug-coded mortality by sex and age group (see S1 Table for the full model results). In Model 1, where lung cancer mortality is omitted, the coefficients are generally high between ages 25 and 59, although the age pattern differs by sex. Among men, the coefficient peaks in the late 50s, with a second, smaller peak at ages 30–34. In women, there is a much larger peak at ages 30–34, with a secondary peak in the early 50s, although the coefficient is relatively stable throughout ages 35–54.

When we add lung cancer mortality in Model 2, estimating the association with drug use and smoking simultaneously, there is little change in the estimated drug coefficients at younger ages. However, at older ages, especially ages 55–64, the drug coefficient is larger after accounting for smoking. For example, at ages 60–64, the drug coefficient for men increases from 0.32 in Model 1 to 0.52 in Model 2; among women, the negative coefficient in Model 1 (-0.08) becomes positive in Model 2 (0.21). The addition of lung cancer improves model fit based on BIC. These results suggest that the model that omits smoking under-estimates the association with drug use, perhaps because of the negative correlation over time and space between lung cancer and drug-coded mortality, especially for people aged 55–74 (results not shown). Thus, for the remainder of the analysis, we focus on results from the model that includes smoking. In Model 1 that omits lung cancer, the estimated number of drug-associated deaths at ages 15–64 is 128,040 (results not shown). This number is 9.6% lower than the estimate based on Model 2 that includes lung cancer.

Our estimates suggest that, between ages 15 and 64, drug use cost more than 140,000 American lives in 2016 (98,508 men and 43,187 women; Table 1). The percentage of all-cause deaths at ages 15–64 in 2016 that are estimated to be drug-associated is 22% for men and 16% for women. These values are more than double the fraction coded directly to drug use (9% and 8%, respectively; Table 1). At older ages, the ratio is even higher; for example, among men ages 55–59, the drug-associated fraction (22%) is more than five times the drug-coded percentage (4%); the ratio is 3.5 among their female counterparts (14% vs. 4%, respectively).

The drug-associated fraction depends not only on the estimated drug coefficient, but also on the observed level of drug-coded mortality in that subgroup (see Eq 3 in S3 Appendix). Therefore, although the drug coefficient at ages 30–34 is more than twice as high for women as for men, the fraction of drug-associated deaths is slightly lower in women (e.g., 38% for women vs. 41% for men in 2016; Table 1) because drug-coded mortality rates are lower in women.

The specification for the time trend has relatively little effect on the results. The estimated number of drug-associated deaths at ages 15–64 is only 3% lower (137,908, not shown) when we substitute dummies for each year in place of the linear term.

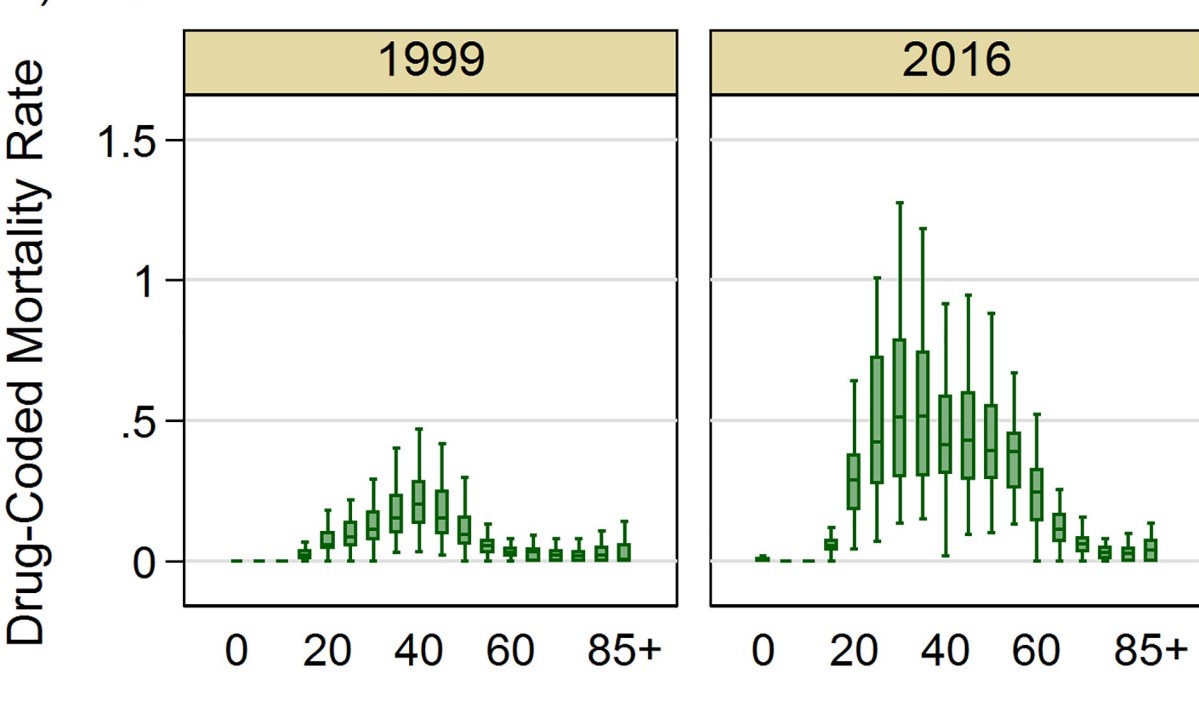

**Fig 1. Box plot of drug-coded mortality rate (per 1,000) among 50 US states, by five-year age group in 1999 and 2016 among A) males and B) females.** Note: In these plots, the box represents the 25th (p25), 50th (p50), and 75th (p75) percentiles of the distribution; the whiskers represent the lower and upper adjacent values. The lower adjacent value is defined as p25−1.5×IQR, where IQR = p75−p25. Similarly, the upper adjacent value is defined as p75+1.5×IQR. Values outside of the whiskers are not shown.

Cause-specific models also produce similar results (see S5 Appendix for details). The estimated number of drug-associated deaths at ages 15–64 in 2016 is 3.6% higher (146,849) than the estimate based on the model that combined all other causes (141,695; S2 Table). As shown in Fig 2, drug-coded mortality accounts for the largest share of drug-associated deaths (41% among men, 46% among women), while the remainder come primarily from circulatory diseases, other external causes, non-lung cancers, digestive diseases, and respiratory diseases.

### State variation in drug-associated mortality

Drug-associated mortality varies substantially by state. Among deaths at ages 15–64 in 2016 (Table 2), the drug-associated fraction is highest in West Virginia at 39% for men and 27% for women and lowest in Nebraska (8% and 9%, respectively). Although the fraction for midlife men in West Virginia seems high, relatively few people die at these ages. Among all deaths at ages 15 and older in West Virginia in 2016, the estimated drug-associated percentage was 13% for men and 4% for women (not shown).

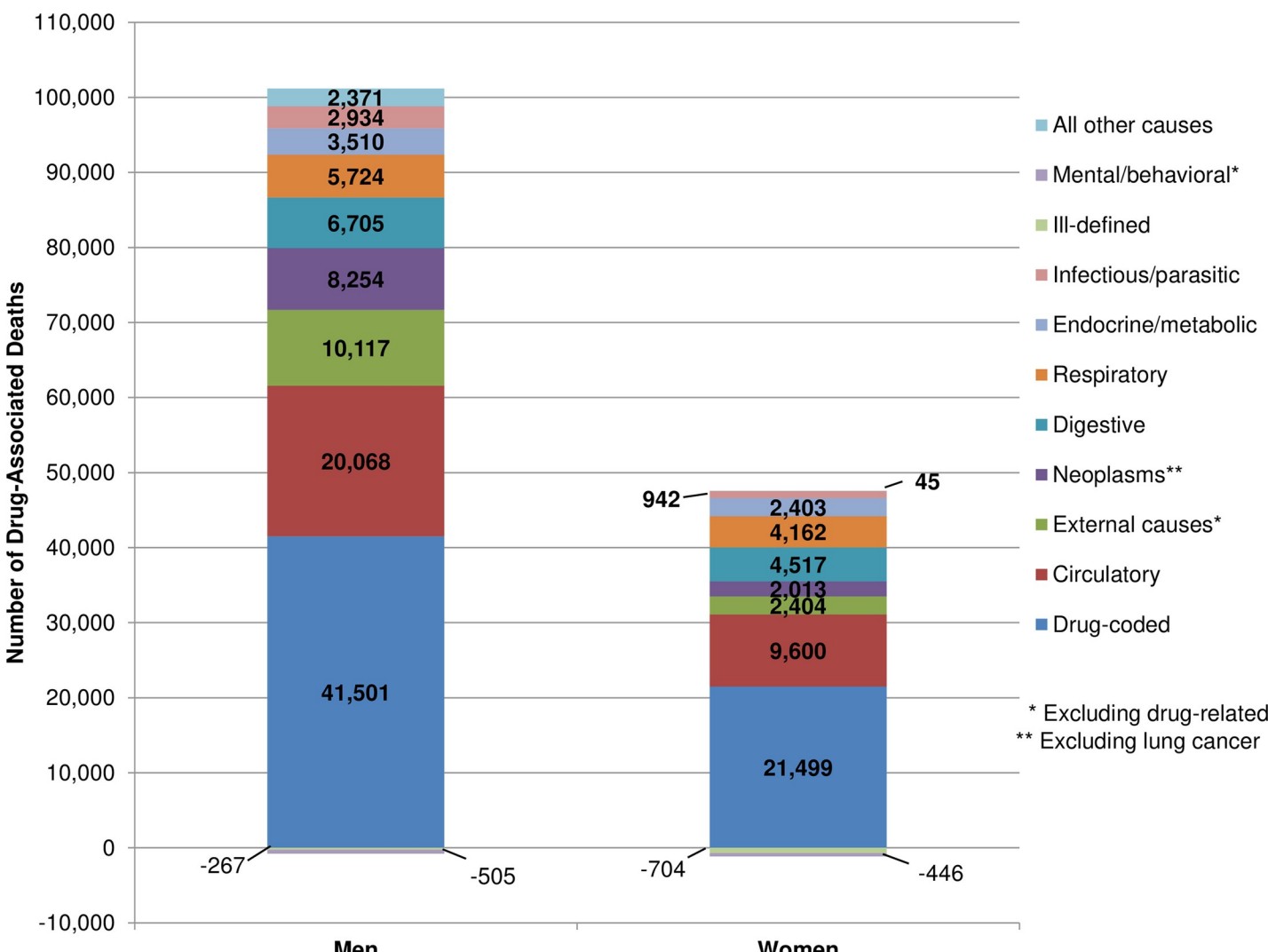

**Fig 2. Estimated number of drug-associated deaths at ages 15–64 across groups of causes, by sex, US, 2016.** Note: See S5 Appendix for ICD-10 codes that define each cause group. Negative values result when the model indicates an inverse association between drug-coded mortality and mortality from the specified cause group.

**Table 2. Percentage of deaths at ages 15–64 that are drug-coded and drug-associated, by sex and state, 2016.**

| State | Drug-Coded | | Drug-associated[a] | |
|---|---|---|---|---|
| | Men | Women | Men | Women |
| Nebraska | 3.1 | 4.3 | *7.6* | *9.1* |
| South Dakota | 3.5 | 4.6 | *7.8* | *9.5* |
| Montana | 4.3 | 6.6 | *10.3* | 15.3 |
| North Dakota | 5.5 | 5.5 | *11.8* | *11.0* |
| Iowa | 5.0 | 4.8 | *11.8* | *9.7* |
| Mississippi | 3.6 | 3.7 | 12.0 | *10.6* |
| Kansas | 4.7 | 5.6 | 12.3 | *11.7* |
| Arkansas | 4.1 | 5.1 | 12.6 | 13.2 |
| Wyoming | 5.6 | 9.1 | 13.0 | 18.3 |
| Texas | 5.0 | 4.7 | 13.4 | *10.0* |
| Alabama | 4.9 | 4.9 | 13.7 | 12.2 |
| Minnesota | 7.9 | 6.4 | 15.3 | *11.1* |
| Georgia | 5.4 | 5.2 | 15.3 | *11.8* |
| Vermont | 9.2 | 9.3 | 15.7 | 16.4 |
| Idaho | 7.4 | 7.9 | 16.2 | 16.0 |
| Oregon | 6.4 | 6.2 | 16.7 | 12.4 |
| South Carolina | 6.3 | 6.2 | 18.2 | 14.7 |
| Alaska | 7.5 | 7.3 | 18.2 | 16.0 |
| North Carolina | 7.9 | 7.9 | 18.5 | 16.0 |
| Virginia | 8.7 | 7.0 | 18.6 | 13.2 |
| California | 7.2 | 6.4 | 18.8 | 12.7 |
| Wisconsin | 9.1 | 8.8 | 18.9 | 16.1 |
| Colorado | 8.7 | 9.5 | 19.7 | 17.2 |
| Louisiana | 7.0 | 7.2 | **20.5** | 16.9 |
| Washington | 8.3 | 8.5 | **20.6** | 16.1 |
| Oklahoma | 6.7 | 7.3 | **21.4** | 18.4 |
| Indiana | 9.1 | 8.7 | **21.5** | 18.0 |
| Missouri | 8.8 | 8.3 | **21.5** | 17.2 |
| Illinois | 9.9 | 7.2 | **22.8** | 13.7 |
| Tennessee | 8.1 | 8.8 | **23.2** | **21.3** |
| Hawaii | 8.4 | 5.9 | **24.0** | 12.3 |
| New Mexico | 8.9 | 9.4 | **24.2** | **21.2** |
| Nevada | 8.6 | 9.7 | **24.3** | **21.4** |
| Florida | 10.4 | 8.7 | **24.8** | 17.6 |
| Maine | 11.9 | 10.2 | **24.9** | 17.9 |
| Arizona | 9.8 | 8.6 | **25.2** | 17.6 |
| New York | 11.9 | 7.7 | **25.3** | 13.5 |
| Michigan | 11.1 | 9.5 | **26.5** | 19.2 |
| Utah | 12.6 | 13.6 | **26.6** | **24.9** |
| New Jersey | 13.8 | 9.2 | **27.4** | 16.0 |
| Kentucky | 10.6 | 9.9 | **28.5** | **22.9** |
| Delaware | 14.2 | 10.6 | **30.4** | 20.1 |
| New Hampshire | 17.5 | 13.6 | **32.1** | 22.0 |
| Rhode Island | 16.8 | 11.6 | **33.7** | 22.0 |
| Connecticut | 16.7 | 11.2 | **34.0** | 19.2 |
| Ohio | 15.0 | 11.9 | **34.1** | 23.7 |

*(Continued)*

**Table 2.** (Continued)

| State | Drug-Coded | | Drug-associated[a] | |
|---|---|---|---|---|
| | Men | Women | Men | Women |
| Pennsylvania | 16.1 | 12.8 | **34.4** | **23.2** |
| Maryland | 17.0 | 11.2 | **38.6** | **20.9** |
| Massachusetts | 20.8 | 14.3 | **38.7** | **23.2** |
| West Virginia | 14.9 | 12.0 | **38.7** | **26.8** |

[a] Includes drug-coded deaths as well as deaths from other causes (i.e., underlying cause was not drugs or lung cancer) estimated to be drug-associated based on Model 2. Values greater than 20% are shown in **bold** type. Values less than 12% are shown in *italics*.

The drug epidemic appears to have left much of the Plains states relatively unscathed, including Nebraska, Iowa, Kansas, and the Dakotas. Regions of high drug mortality are more dispersed. They include Appalachia (West Virginia, Kentucky, and Tennessee), parts of New England, Pennsylvania, Maryland, Delaware, Ohio, and much of the Southwest (Arizona, New Mexico, Nevada, and Utah).

### Estimating life expectancy and survival probabilities in the absence of drug use

Fig 3 shows the growing impact of drug use on life expectancy at age 15. By 2016, we estimate that drug-associated mortality cost men 1.4 years and women 0.7 of a year of life (S3 Table). Without the imprint of drug use on national mortality levels, life expectancy at age 15 would have increased slightly between 2014 and 2016 (from 63.5 to 63.6 years among men and from 67.8 to 67.9 years among women) instead of declining. Between 2010 and 2016, life expectancy at age 15 would have improved by 0.5 years (from 63.1 to 63.6) among men and by 0.4 of a year among women (from 67.5 to 67.9 years) instead of stagnating.

When analyzed separately by state (S3 Table), the loss of life years associated with drug use ranged from 0.4 (Nebraska) to 3.6 (West Virginia) for men and from 0.3 (Nebraska & Iowa) to 1.9 (West Virginia) for women. The gap in observed life expectancy at age 15 in 2016 between men in Nebraska (63.4) and West Virginia (58.1) was 5.3 years; in the absence of drug use, we estimate that the gap would have been only 2.1 years (63.8 vs. 61.7, respectively; results not shown).

The percentage of the population at age 15 that is expected to die by age 65 is shown in Fig 4. The probability of dying in midlife rose between 2010 and 2016, from 18.4% to 19.4% among men and from 11.2% to 11.8% among women. Our estimates imply that such an increase is largely attributable to drug use. In the absence of drug use, the predicted percentage dying in midlife would have decreased over that period among men (from 16.2% to 15.4%) and increased only slightly among women (from 9.9% to 10.0%). In 2016, we estimate that an additional 3.9% of men and 1.8% of women died between ages 15 and 65 because of drug use (S3 Table). Drug use was responsible for a lot more excess midlife mortality in some states (e.g., 9.8% of men and 4.6% of women in West Virginia) than in others (e.g., about 1% in Nebraska, S3 Table).

### Discussion

A recent study estimated that increases in drug poisoning mortality between 2000 and 2015 resulted in a 0.28 year loss in life expectancy at birth [33]. Another study reported that drug overdose accounted for 42% of the 2014–15 decline in life expectancy among men and 18% of

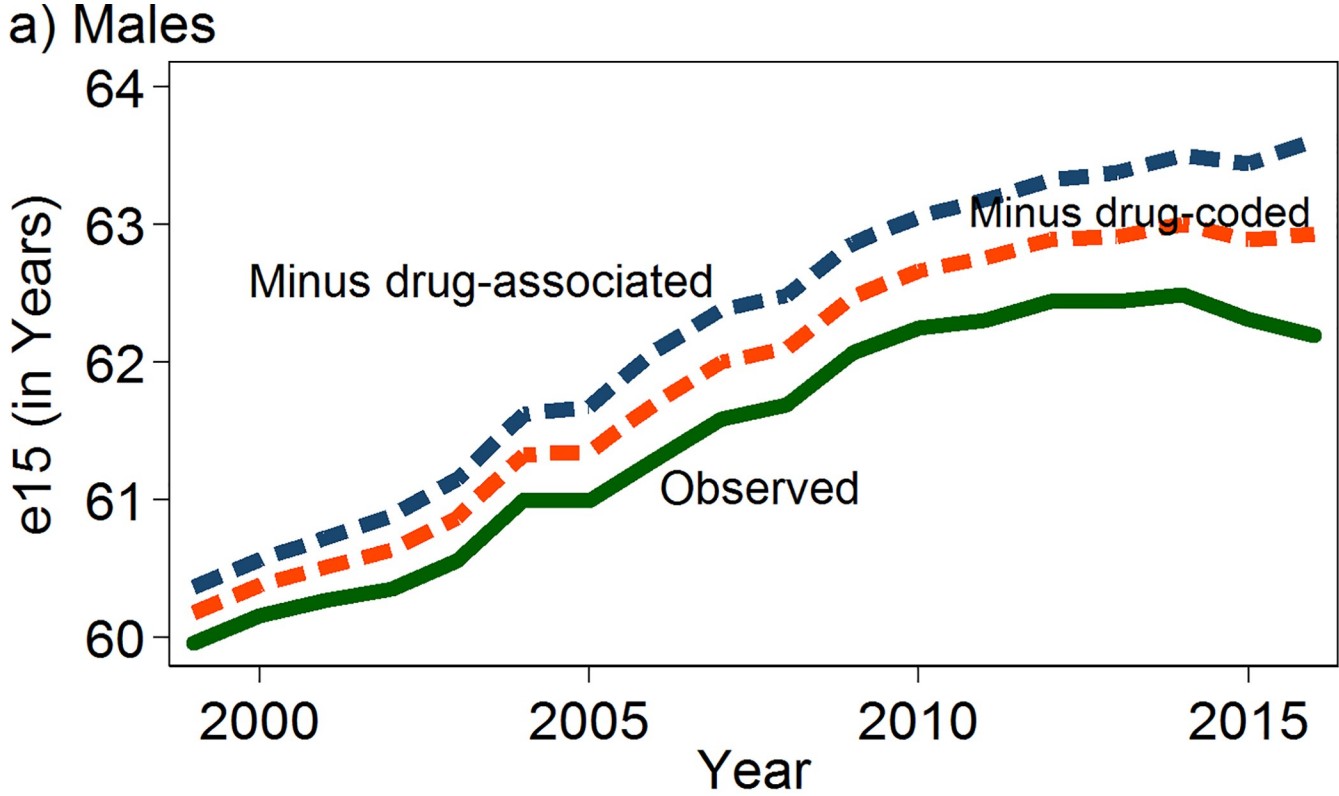

a) Males

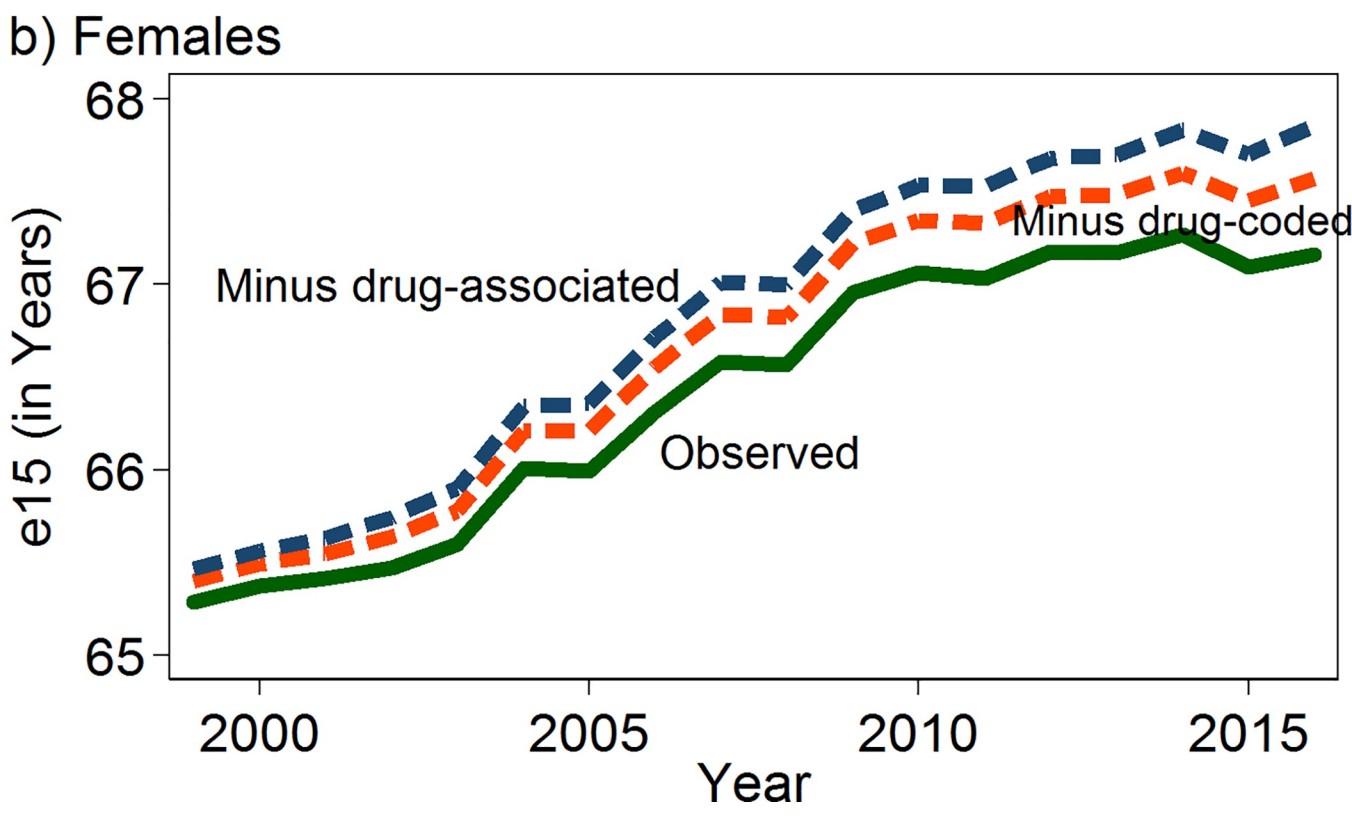

b) Females

**Fig 3. Life expectancy at age 15 ($e_{15}$) with and without drug use, US, 1999–2016, by sex.** Note: Estimates are based on the model that includes smoking.

the decline in women in the US [34]. Other research demonstrated that increased drug mortality accounts for all of the rise in mortality rates between 1999 and 2015 among non-Latino whites aged 22–56 [35]. Researchers have also evaluated whether drug mortality accounts for the growing educational gap in life expectancy [36, 37], but none of these studies accounted for excess mortality from other causes of deaths that might have resulted from drug use. Among Americans aged 15–64 in 2016, we estimate that drug use cost 141,695 lives—2.2 times the number of drug-coded deaths (63,000). In terms of life expectancy beyond age 15, we find that drug use cost men 1.4 years and women 0.7 years, on average. In the hardest-hit state (West Virginia), drug use cost men 3.6 and women 1.9 life years. To our knowledge, only one other study has attempted to estimate the full extent of drug-associated mortality. Using a different method (i.e., adding deaths for which drugs were coded as a contributing cause) and providing estimates only for Finland [8], they reported that the total number of drug-associated deaths at ages 15–64 was 2.8 times the number of drug-coded deaths (as the underlying cause), a ratio that is higher than what we estimate for the U.S.

Our analysis estimates the magnitude of the drug problem, but does not explain the root causes of the drug epidemic. Currently, there are two main types of hypotheses to explain the recent US mortality crisis. One type implicates supply factors, especially an increased availability of opioids [38–42]. An alternative hypothesis implicates demand factors, in particular industrial changes that have generated a rising tide of despair especially among less-educated, middle-aged Americans [43–47]. Case and Deaton [43, 44] suggested that increasing midlife distress may explain increased death rates among middle-aged non-Hispanic whites not only from drug overdose but also from suicide and alcohol-related mortality. Our estimates of drug-associated mortality would be biased upwards if an underlying factor of despair is causing increases in both drug-coded mortality and death rates from other causes, producing a spurious correlation.

To investigate the potential magnitude of this bias, we examined interstate correlations between changes from 1999 to 2016 in drug-coded, alcohol-related, and non-drug suicide mortality rates for each sex/age group. Results are shown in S4 Table. A majority of the age and sex-specific correlations between drug-coded mortality and either alcohol or suicide mortality are negative and very few of the positive values exceed 0.2. We conclude that joint dependence of drug-coded deaths, alcohol deaths and suicide on a common factor of despair is not likely to be an important source of bias in our estimates of drug-associated mortality.

One limitation of our study is that interstate variation in cause of death reporting practices may affect our results. Jalal et al. [48] notes that the practice of identifying, or capacity to determine, intent for drug poisonings varies across states. To minimize this problem we defined drug-coded deaths to include all drug poisonings, regardless of intent, as well as drug-coded mental disorders. Nonetheless, there may still be interstate variation in the likelihood of reporting a death as attributable to drug use, which would, by itself, create a negative association between drug-coded deaths and deaths from other causes. In turn, this correlation would reduce the coefficients relating drug-coded deaths to deaths from other causes and reduce the estimated number of deaths indirectly associated with drugs. In cause-specific models, we find a negative relationship between drug-coded mortality and deaths from two other groups of causes: ill-defined causes and non-drug-related mental and behavioral disorders (see S5 Appendix). We speculate that some drug-coded deaths may be miscoded to ill-defined causes or to non-drug-related mental and behavioral disorders, thereby enhancing the likelihood that we have underestimated drug-associated mortality. Below age 45, there is also a negative

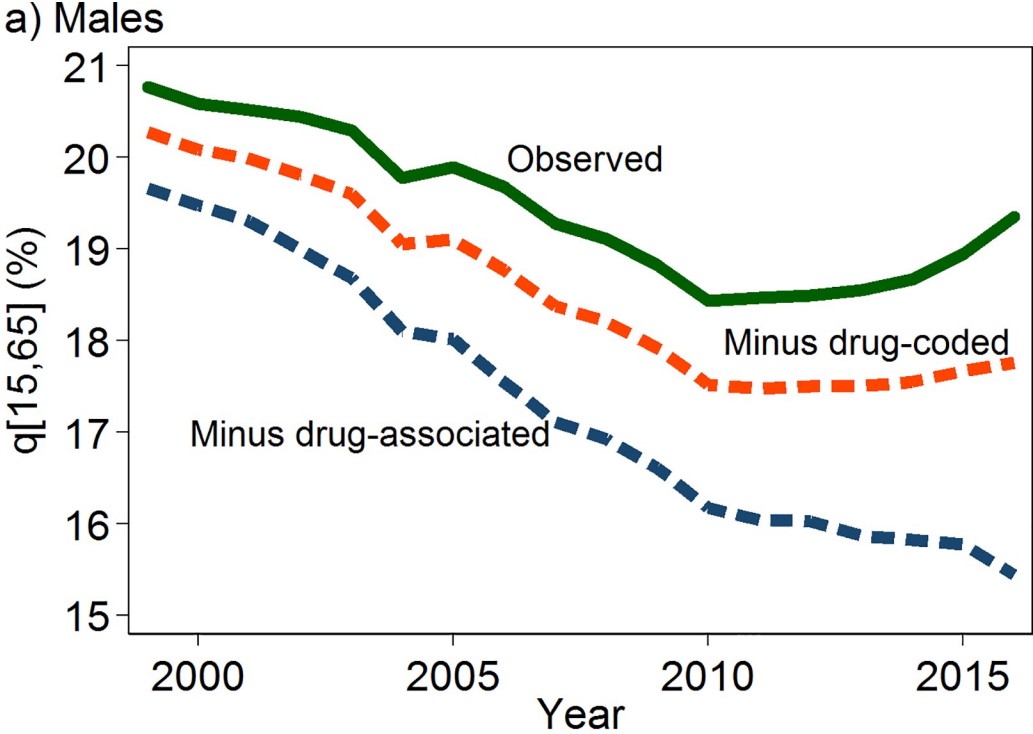

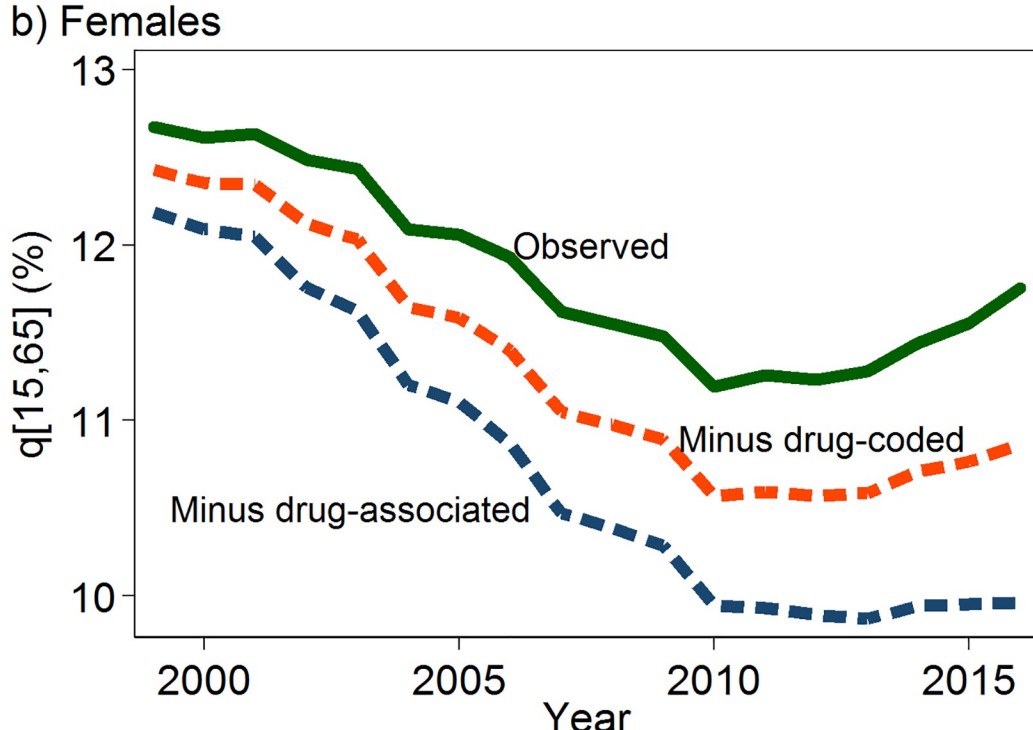

**Fig 4. Percentage dying between age 15 and 65 ($q$[15,65]) with and without drug use, US, 1999–2016, by sex.** Note: Estimates are based on the model that includes smoking.

association with infectious/parasitic diseases, which could result from misclassification between drug-related causes and HIV/hepatitis.

Second, there may be state-level variation in policies to prevent overdose death (e.g., use of naloxone), which would compromise our assumption that geographic differences in drug-coded mortality result solely from differences in drug use. If such extraneous factors are operating, they may affect the relationship between drug-coded mortality and other drug-associated deaths. Geographic variation in the composition of drug type usage may also influence the cause of death structure.

A third limitation is that our estimates of drug-associated mortality cannot distinguish between deaths to drug users and secondhand effects (e.g., increased mortality among non-drug users because high drug use within a community may facilitate the spread of infectious disease, increase drug-related crime, and heighten exposure to interpersonal strains, violence, and abuse within families). Because we model relations between drug-coded mortality rate and deaths from other causes within the same sex and age group, our model is not likely to capture secondhand effects unless they occur within the same sex and age group. Secondhand effects are likely to be more diffuse, influencing people both younger and older than drug users and of the opposite sex (e.g., spouses/partners). For example, drug-coded mortality rates below age five increased by more than 60% between 1999 and 2016, probably as a result of pre-natal drug use by the mother. If secondhand effects of drug use are notable, then again, we may have under-estimated the impact of drug use on overall mortality.

Fourth, the time lag between drug use and other causes of death may be different than for drug-coded mortality. For example, overdose occurs within a short period after drug use, whereas deaths resulting from myocardial infarction or liver damage could happen long after drug use ceases. Our model would not effectively capture the long-term consequences of drug use.

A final limitation, common to nearly all observational studies, is that results may be biased by the omission of important variables. Variables that predict both drug-associated mortality and mortality from causes unrelated to drug use may bias our coefficients and resulting drug-associated fractions. For example, chronic illnesses could increase pain levels, which in turn might lead to opioid use and abuse. Thus, there could be a spurious association between over-dose deaths and mortality from chronic diseases like heart disease and diabetes. We have endeavored to capture the aggregate effect of many of these omitted variables by distinguishing age and sex, by including state fixed effects and time trends, and by controlling for the impact of smoking. If we have not succeeded in representing the role of important omitted variables successfully through these proxies, our coefficients may be biased. What sets our study apart from many other observational studies is that the process whose influences we seek to identify has shown remarkable variation over time and space. Background variables that influence mortality (e.g., educational attainment, obesity, and medical technology) operate much more slowly and systematically over time and space. They should not be highly correlated with the powerful dynamics of drug use.

## Conclusion

While it is clear that the drug epidemic has taken a heavy toll in America, the full impact of the drug epidemic on US mortality is substantially higher than deaths resulting directly from over-dose would suggest. Using drug-coded deaths as an indicator of the dynamics of drug use, we conclude that drug use has left an imprint on US mortality that is roughly double that implied by drug-coded deaths alone. Our results suggest that drug-associated mortality largely accounts for adverse trends since 2010 in midlife mortality. In the absence of drug use, we

estimate that the probability of dying between ages 15 and 65 would have continued to decline after 2010 among men (from 16.2% in 2010 to 15.4% in 2016) and would have remained at a low level (9.9% in 2010, 10.0% in 2016) among women. Recent declines in US life expectancy have been blamed largely on the drug epidemic [49–52]. Our estimates indicate that, in the absence of drug use, life expectancy at age 15 would have increased slightly between 2014 and 2016. The drug epidemic is exacting a heavy cost to American lives, not only from deaths directly coded to drugs but also from excess mortality in other causes of death affected by drug use.

## Supporting information

**S1 Appendix. Modeling the association between drug-coded mortality and mortality from all other causes of death.**
(DOCX)

**S2 Appendix. Modeling the association between drug-coded mortality, smoking mortality, and mortality from all other causes of death.**
(DOCX)

**S3 Appendix. Estimating the drug-associated fraction.**
(DOCX)

**S4 Appendix. Estimating life expectancy at age 15 in the absence of drug use.**
(DOCX)

**S5 Appendix. Cause-specific models.**
(DOCX)

**S1 Fig. Age-standardized drug-coded mortality rate (per 1,000) for the US, West Virginia, and Nebraska, ages 15 and older, 1981–2016.**
(DOCX)

**S1 Table. Estimated coefficients from negative binomial regression using drug-coded mortality rate as a predictor, 1999–2016, modeled separately by sex.**
(DOCX)

**S2 Table. Estimated number of drug-associated deaths among those aged 15–64 and distribution across groups of causes by sex, US, 2016.**
(DOCX)

**S3 Table. Estimated difference in life expectancy at age 15 ($e_{15}$) and percentage dying between age 15 and 65 ($q[15,65]$) associated with drug use, by state and sex, 2016.**
(DOCX)

**S4 Table. Interstate correlations between changes (2016–1999) in drug-coded mortality and corresponding changes in alcohol-related mortality and non-drug suicide rates by age and sex.**
(DOCX)

## Acknowledgments

We are grateful to Maxine Weinstein and Jessica Ho for their comments on the manuscript.

## Author Contributions

**Conceptualization:** Dana A. Glei, Samuel H. Preston.

**Data curation:** Dana A. Glei.

**Formal analysis:** Dana A. Glei.

**Funding acquisition:** Samuel H. Preston.

**Investigation:** Dana A. Glei, Samuel H. Preston.

**Methodology:** Dana A. Glei, Samuel H. Preston.

**Visualization:** Dana A. Glei.

**Writing – original draft:** Dana A. Glei.

**Writing – review & editing:** Dana A. Glei, Samuel H. Preston.

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
