## [Decision Letter · Decision Letter 0]

25 Oct 2019

PONE-D-19-25601

Estimating the impact of drug use on US mortality, 1999-2016

PLOS ONE

Dear Dr. Glei,

Thank you for submitting your manuscript to PLOS ONE. After careful consideration, we feel that it has merit but does not fully meet PLOS ONE’s publication criteria as it currently stands. Therefore, we invite you to submit a revised version of the manuscript that addresses the points raised during the review process.

Reviewers highlighted the relevance and timeliness of this manuscript. However, they identified some minor issues that require your attention before this paper can be published.

We would appreciate receiving your revised manuscript by Dec 09 2019 11:59PM. To enhance the reproducibility of your results, we recommend that if applicable you deposit your laboratory protocols in protocols.io, where a protocol can be assigned its own identifier (DOI) such that it can be cited independently in the future. For instructions see: http://journals.plos.org/plosone/s/submission-guidelines#loc-laboratory-protocols

We look forward to receiving your revised manuscript.

Kind regards,

João Pedro Silva, PhD

Academic Editor

PLOS ONE

Journal Requirements:

2)  We note that you have indicated that data from this study are available upon request. PLOS only allows data to be available upon request if there are legal or ethical restrictions on sharing data publicly. For more information on unacceptable data access restrictions, please see http://journals.plos.org/plosone/s/data-availability#loc-unacceptable-data-access-restrictions.

Reviewers' comments:

Reviewer's Responses to Questions

**Comments to the Author**

1. Is the manuscript technically sound, and do the data support the conclusions?

Reviewer #1: Yes

Reviewer #2: Yes

2. Has the statistical analysis been performed appropriately and rigorously? 

Reviewer #1: Yes

Reviewer #2: Yes

3. Have the authors made all data underlying the findings in their manuscript fully available?

Reviewer #1: Yes

Reviewer #2: Yes

4. Is the manuscript presented in an intelligible fashion and written in standard English?

Reviewer #1: Yes

Reviewer #2: Yes

5. Review Comments to the Author

Reviewer #1: Overall, the study is an excellent complement to the Global Burden of Disease (GBD) Study in estimating the drug-coded related mortality. It provided a robust estimate of the current condition of the drug-use epidemic in the United States in the broader year period from 1999-2016. The reviewer would want to point out some minor issues which the reviewer thinks might enhance the quality of the publication.

1. In the introduction, the authors used the term "drug-use" repeatedly without an effort to elaborate on the definition further. The reviewer understood that they had used the ICD-10 definition, which the reviewer only found out in the methods section. Thus, the reviewer suggests defining the scope of the term "drug-use" earlier in the introduction section.

2. In lines 46-49. The authors stated the following

46 "obvious connection with accidental poisoning, drug use may affect other disease and injury

47 processes resulting in deaths assigned to other causes that would not necessarily be

48 considered drug-related. A meta-analysis indicated that standardized mortality rates among

49 opioid-dependent individuals are almost 15 times those of the general population."

What is not clear for the reviewer is when they refer to other "diseases and injury not necessarily be considered drug-related. The reviewer understood that for these conditions, drug-use might be an antecedent cause of maybe a contributory or precipitating factors that led to these outcomes.

3.The authors mentioned in line 58 "viral hepatitis." Viral hepatitis has several subtypes; the reviewer suggests to specify the subtype (B, C), etc.

4.In line 65, the authors repeated that drugs are an indirect cause which the reviewer finds confusing

5.Drug use definition, this can be resolved when the suggestion number 1 is addressed.

6.There are related articles such as the one from Shiels https://www.thelancet.com/journals/lanpub/article/PIIS2468-2667(18)30208-1/fulltext and Ruhm https://pophealthmetrics.biomedcentral.com/articles/10.1186/s12963-016-0071-7 which have used the same data sources but was not cited in the introduction. The reviewer suggests referencing the previous studies and how different the current study is when compared to both.

METHODS:

The reviewer would want to see an explanation on the omission of alcohol in the elaboration model such as mortality from drugs vis-a-vis alcohol use vs drugs plus alcohol use. The authors did a model elaboration for smoking but not for alcohol consumption. A good reference is included below.

The second component of the key indicator complements routine statistics and provides data on the overall and cause specific mortality rate based on a cohort of drug users, usually in contact with drug treatment services. The mortality rates include deaths directly induced by use of drugs (overdoses) and deaths that may be indirectly related to drug use such as infectious diseases, injuries and violence, suicides, and other causes of death that may be related to other aspects (such as smoking or alcohol-related causes, mental health problems, or social exclusion). However, this component is less readily available in all countries as it is based on specific studies, which are more resource intensive and often limited in time and geographical coverage, compared to the first component of the fundamental indicator.

http://www.emcdda.europa.eu/attachements.cfm/att_67050_EN_EMCDDA-DRD-overview.pdf

2. The authors mentioned at line 195 that they anticipate that non-drug related

196 mental/behavioural disorders and ill-defined causes may be negatively associated with drug197

coded mortality because these are the categories where coding confusions are most likely to

198 occur with drug-coded mortality. The reviewer would want to know the reference for this assumption.

DISCUSSION:

The discussion should also discuss how the findings are in contrary or how it supports the earlier findings of the previous studies that address the same objectives. Both of the previous studies the reviewer referenced here noted mixed results where there is a notable decline in subgroups, but there is a remarkable rise in other subgroups.

Reviewer #2: This paper quantifies the mortality burden from narcotic drug use in the US. The paper is timely: the number of drug-coded deaths in the US is alarmingly high. However not all drug-related deaths are coded as such. Some of these deaths are due to the sequelae of narcotics use, and might include for example cardiovascular related deaths from cocaine use. The current paper uses a regression-based method to estimate the number of narcotic related deaths that are not coded as such. In particular, the authors examine the association between drug-coded mortality rates (MD) and death rates from all other causes (MND) for the US population aged 15 and older. The models are estimated using mortality data collected for each of the 27,000 unique combinations of US state, sex, 5-year age group and year over the period 1999-2016. More specifically, the authors estimate negative binomial regressions of the log of MND as a function of MD, indicators of state, sex, age group and year, and interactions between MD and age group indicators (so as to allow the impact of MD on MND to vary by age). After estimating the model parameters, the authors predict log MND given a value of MD (and presumably the values of the indicator variables). The authors then estimate a number of additional subgroup specific models. These variations include models of cause-specific MND.

The results are striking. Among non-seniors, there were 2.2 MND deaths for every MD. Most of these MND deaths were circulatory related, as one might expect.

I was impressed with the overall quality of the study. It appears to be the first study of its kind for the US. It was well written, thorough and the limitations were clearly expressed. The most important limitation is that of omitted variables bias – these are variables that cause MND that are related to MD. The authors give the following example:

“For example, chronic illnesses could increase pain levels, which in turn might lead to opioid use and abuse. Thus, there could be a spurious association between overdose deaths and mortality from chronic diseases like heart disease and diabetes.”

The authors also indicate another example:

“Case and Deaton [39, 40] suggested that increasing midlife distress may explain increased death rates among middle-aged non-Hispanic whites not only from drug overdose but also from suicide and alcohol-related mortality. Our estimates of drug associated mortality would be biased upwards if an underlying factor of despair is causing increases in both drug-coded mortality and death rates from other causes, producing a spurious correlation.”

I wonder if the problem is as severe as the authors indicate. In particular, wouldn’t the time indicators pick up these rising levels of midlife crises?

If not, then the authors rightly interpret their estimates as an upper bound on the estimated impact of MD on MND.

I had only one substantive suggestion for the authors. I wonder if it is possible to use instrumental variables techniques here – this would require instruments, variables strongly associated with MD that are not associated with the confounding factors that the authors allude to. Is the case for instance that the local availability of narcotics (being close to international smuggling routes) affects MD. If so then this variable might be profitably used to implement an IV strategy. Certainly the sample size is large enough to do so.

6. PLOS authors have the option to publish the peer review history of their article (what does this mean?). If published, this will include your full peer review and any attached files.

Reviewer #1: Yes: MELVIN BARRIENTOS MARZAN

Reviewer #2: Yes: Paul Grootendorst

---

## [Author Response · Author response to Decision Letter 0]

12 Nov 2019

Reviewer #1

Overall, the study is an excellent complement to the Global Burden of Disease (GBD) Study in estimating the drug-coded related mortality. It provided a robust estimate of the current condition of the drug-use epidemic in the United States in the broader year period from 1999-2016. The reviewer would want to point out some minor issues which the reviewer thinks might enhance the quality of the publication.

1. In the introduction, the authors used the term "drug-use" repeatedly without an effort to elaborate on the definition further. The reviewer understood that they had used the ICD-10 definition, which the reviewer only found out in the methods section. Thus, the reviewer suggests defining the scope of the term "drug-use" earlier in the introduction section.

RESPONSE: We have added the following sentence to the third paragraph of the paper to clarify that we are using drug-coded mortality as an indicator of drug use: “In this study, we use annual death rates by state to model the relationship between drug-coded mortality—which serves as our indicator of the population-level prevalence of drug use—and mortality from other causes.” 

We have also added the following additional sentence to clarify the definition of drug-coded mortality and the broad range of drugs it covers: “Drug-coded mortality includes deaths from all drugs, medicaments, and biological substances (e.g., opioids, cannabinoids, sedatives/hypnotics, cocaine, other stimulants, hallucinogens, volatile solvents, and other psychoactive substances).” While most of those deaths are likely to reflect drug misuse or abuse, some may result from prescribed use of prescription drugs with unanticipated consequences.

2. In lines 46-49. The authors stated the following

46 "obvious connection with accidental poisoning, drug use may affect other disease and injury

47 processes resulting in deaths assigned to other causes that would not necessarily be

48 considered drug-related. A meta-analysis indicated that standardized mortality rates among

49 opioid-dependent individuals are almost 15 times those of the general population."

What is not clear for the reviewer is when they refer to other "diseases and injury not necessarily be considered drug-related. The reviewer understood that for these conditions, drug-use might be an antecedent cause of maybe a contributory or precipitating factors that led to these outcomes.

RESPONSE: Thank you. We have clarified the sentence in the following way: “In addition to the obvious connection with accidental poisoning, drug use may increase the risk of dying from other disease and injury processes in ways that are not recognized in assignments of underlying or even contributing cause of death.”

3.The authors mentioned in line 58 "viral hepatitis." Viral hepatitis has several subtypes; the reviewer suggests to specify the subtype (B, C), etc.

RESPONSE: Our first mention of “viral hepatitis” was on line 51 on the submission and it was mentioned again on line 54 (the page numbers on the reviewers’ version are apparently different than on the version we submitted). In both cases, the studies we cite did not specify a particular type of hepatitis nor did they specify the ICD codes they used to define “viral hepatitis.” Thus, we cannot presume the types of hepatitis to which the authors were referring. Later in the manuscript (lines 82-85 of the original submission), we stated that, “The effects of drug use on the prefrontal circuits in the brain can impair judgement, thereby increasing risky behavior (e.g., driving under the influence, unprotected sex, needle/syringe sharing) [14] that indirectly heightens risk of accidents, injury, trauma, and infectious disease (e.g., HIV/AIDS, hepatitis) [15].” In this case, we are citing a NIDA webpage (which again does not specify the types of hepatitis). However, we have added a citation to another NIDA webpage (www.drugabuse.gov/related-topics/viral-hepatatis-very-real-consequences/substance-use), which does refer to both hepatitis B and C as a particular risk among drug users. Thus, we have clarified that sentence to indicate that illicit drug use increases the risk of acquiring “viral hepatitis B and C.” 

4.In line 65, the authors repeated that drugs are an indirect cause which the reviewer finds confusing

RESPONSE: Again, we are having difficulty following the reviewer’s line references because they appear to differ from the line numbers in our submission. Line 65 of our submission reads, “These associations suggest that, in addition to its direct effect on deaths from poisoning,...” That is, there is no mention of the word “indirect” in that line or in the remainder of that sentence. On line 61-62, we stated, “Among the deaths indirectly associated with drugs,...” Perhaps that is the line to which the reviewer is referring? Or perhaps the reviewer means the line we mentioned in the response to point #3 above (which appeared on lines 82-85 of the submission)? For the first of those mentions, we have modified the sentence to read, “Among the deaths in which drugs were a contributing cause,...” In the case of the second mention, we have eliminated the word “indirectly” from the sentence. 

5.Drug use definition, this can be resolved when the suggestion number 1 is addressed.

RESPONSE: As noted in our response to point #1, we have added a sentence in the Introduction to clarify that we are using drug-coded mortality as an indicator of drug use.

6.There are related articles such as the one from Shiels https://www.thelancet.com/journals/lanpub/article/PIIS2468-2667(18)30208-1/fulltext and Ruhm https://pophealthmetrics.biomedcentral.com/articles/10.1186/s12963-016-0071-7 which have used the same data sources but was not cited in the introduction. The reviewer suggests referencing the previous studies and how different the current study is when compared to both.

RESPONSE: We have added a reference to the Shiels et al. paper in the introduction. As the reivewer noted, Shiels et al. use the same data that we use from the National Center for Health Statistics but their data are unadjusted (i.e., they count only deaths for which drug poisoning was the underlying cause) and ours are adjusted (i.e., we estimate the number of other drug-associated deaths not coded to a drug-related underlying cause). We make comparisons throughout the Results and Discussion section between our results and those such as Shiels et al.’s that rely on unadjusted deaths from drug overdose. The Ruhm paper adjusts the distribution of deaths by type of drug within the category of drug poisoning, in particular reallocating deaths from “unspecified drugs.” However, we are not attempting to quantify the number of deaths resulting from a particular drug. Rather, we are simply interested in the total number of deaths associated with use of all drugs. All the drug fatalities that Ruhm is redistributing are already included in our analysis. So that Ruhm paper is not directly relevant to our work. However, there is another paper by Ruhm [1] that is relevant to our analyses; we have cited it in the Discussion.

METHODS:

The reviewer would want to see an explanation on the omission of alcohol in the elaboration model such as mortality from drugs vis-a-vis alcohol use vs drugs plus alcohol use. The authors did a model elaboration for smoking but not for alcohol consumption. A good reference is included below.

The second component of the key indicator complements routine statistics and provides data on the overall and cause specific mortality rate based on a cohort of drug users, usually in contact with drug treatment services. The mortality rates include deaths directly induced by use of drugs (overdoses) and deaths that may be indirectly related to drug use such as infectious diseases, injuries and violence, suicides, and other causes of death that may be related to other aspects (such as smoking or alcohol-related causes, mental health problems, or social exclusion). However, this component is less readily available in all countries as it is based on specific studies, which are more resource intensive and often limited in time and geographical coverage, compared to the first component of the fundamental indicator.

http://www.emcdda.europa.eu/attachements.cfm/att_67050_EN_EMCDDA-DRD-overview.pdf

RESPONSE: We now cite this useful paper, which makes the same distinction that we are making in the paper between deaths directly attributable to drug overdose and the excess mortality associated with drug use that is not directly attributable to overdose. It also distinguishes, as we do, between data from vital statistics and data from cohort studies. The approach we are using is original and we point out some weaknesses of cohort studies that are not present in our approach. 

The main reason that we added smoking (i.e., lung cancer mortality) to our model is the overwhelming fraction of US deaths, in the neighborhood of 18-24%, that are attributable to smoking. Alcohol abuse accounts for only about a fifth of that number [2]. Nevertheless, we were concerned that our estimates of the impact of drug use could be upwardly biased by a positive association between drug-related mortality and alcohol-related mortality. Based on results shown in Table S4, the correlations between interstate changes in drug-coded mortality and alcohol-related mortality are mainly small and negative. We concluded that joint dependence of drug-coded deaths and alcohol deaths (as well as suicide) on a common factor is not likely to be an important source of bias in our estimates of drug-associated mortality.

2. The authors mentioned at line 195 that they anticipate that non-drug related

196 mental/behavioural disorders and ill-defined causes may be negatively associated with drug197

coded mortality because these are the categories where coding confusions are most likely to

198 occur with drug-coded mortality. The reviewer would want to know the reference for this assumption.

RESPONSE: Although we believe there is good reason to expect some drug-related deaths to be miscoded to ill-defined causes or non-drug related mental/behavioral disorders, we do not have good citations to support that assumption. Therefore, we have eliminated the sentence highlighted by the reviewer and added the following sentences at the end of the paragraph that better reflect the fact that these statements are speculative: “For example, if drug overdose deaths are sometimes miscoded to ill-defined causes, it may create a negative association between drug-coded mortality and ill-defined causes. Given high co-morbidity between mental illness and drug use [31], it is also possible that variation in diagnosis and coding practices could create a negative relationship between drug-coded mortality and mortality from non-drug related mental/behavioral disorders.”

Later, in the Discussion, we say, “In cause-specific models, we find a negative relationship between drug-coded mortality and deaths from two other groups of causes: ill-defined causes and non-drug related mental and behavioral disorders (see eAppendix 5).” We have modified the subsequent sentence to read, “We speculate that some drug-coded deaths may be miscoded to ill-defined causes or to non-drug related mental and behavioral disorders, thereby enhancing the likelihood that we have underestimated drug-associated mortality.”

DISCUSSION:

The discussion should also discuss how the findings are in contrary or how it supports the earlier findings of the previous studies that address the same objectives. Both of the previous studies the reviewer referenced here noted mixed results where there is a notable decline in subgroups, but there is a remarkable rise in other subgroups.

RESPONSE: At the end of the first paragraph of the Discussion, we discussed and compared our results with the only other study with the same objective as our study. Earlier in that paragraph we also mention five other related studies (including a study by Ruhm) and explain how they differ from our own study. As noted in our response to Point #6, the study by Ruhm mentioned earlier by the reviewer is not relevant to our objectives. The Shiels et al. [3] mentioned earlier by the reviewer also has a very different objective than our own study. They focus on how trends in all-cause and drug poisoning mortality vary by racial/ethnic subgroup, socioeconomic status, county, and metropolitan status. We do not seek to identify differences by those subgroups. Our objective is to estimate the overall burden of drug-associated mortality, how it has changed over time, and its effect on life expectancy. The studies by Shiels et al. and Ruhm count only deaths in which the underlying cause is coded as drug poisoning; neither account for excess mortality from other causes of deaths that might have resulted from drug use. 

Reviewer #2

This paper quantifies the mortality burden from narcotic drug use in the US. The paper is timely: the number of drug-coded deaths in the US is alarmingly high. However not all drug-related deaths are coded as such. Some of these deaths are due to the sequelae of narcotics use, and might include for example cardiovascular related deaths from cocaine use. The current paper uses a regression-based method to estimate the number of narcotic related deaths that are not coded as such. In particular, the authors examine the association between drug-coded mortality rates (MD) and death rates from all other causes (MND) for the US population aged 15 and older. The models are estimated using mortality data collected for each of the 27,000 unique combinations of US state, sex, 5-year age group and year over the period 1999-2016. More specifically, the authors estimate negative binomial regressions of the log of MND as a function of MD, indicators of state, sex, age group and year, and interactions between MD and age group indicators (so as to allow the impact of MD on MND to vary by age). After estimating the model parameters, the authors predict log MND given a value of MD (and presumably the values of the indicator variables). The authors then estimate a number of additional subgroup specific models. These variations include models of cause-specific MND.

The results are striking. Among non-seniors, there were 2.2 MND deaths for every MD. Most of these MND deaths were circulatory related, as one might expect.

I was impressed with the overall quality of the study. It appears to be the first study of its kind for the US. It was well written, thorough and the limitations were clearly expressed. The most important limitation is that of omitted variables bias – these are variables that cause MND that are related to MD. The authors give the following example:

“For example, chronic illnesses could increase pain levels, which in turn might lead to opioid use and abuse. Thus, there could be a spurious association between overdose deaths and mortality from chronic diseases like heart disease and diabetes.”

The authors also indicate another example:

“Case and Deaton [39, 40] suggested that increasing midlife distress may explain increased death rates among middle-aged non-Hispanic whites not only from drug overdose but also from suicide and alcohol-related mortality. Our estimates of drug associated mortality would be biased upwards if an underlying factor of despair is causing increases in both drug-coded mortality and death rates from other causes, producing a spurious correlation.”

I wonder if the problem is as severe as the authors indicate. In particular, wouldn’t the time indicators pick up these rising levels of midlife crises? 

If not, then the authors rightly interpret their estimates as an upper bound on the estimated impact of MD on MND.

RESPONSE: In the Discussion, we noted that, “We have endeavored to capture the aggregate effect of many of these [OMITTED] variables by distinguishing age and sex, by including state fixed effects and time trends, and by controlling for the impact of smoking.” We have added the following additional sentence: “If we have not succeeded in representing the role of important omitted variables successfully through these proxies, our coefficients may be biased.” We cannot find any place in the manuscript where we explicitly stated that our estimates represent an upper limit. We do not intend to imply that. As we have attempted to outline in the Discussion, there are some potential problems (e.g., omitted variable bias) that may lead us to overestimate drug-associated mortality, but there are others (e.g., interstate variation in the likelihood of reporting a death as attributable to drug use, secondhand effects of drug use) that could cause us to under-estimate drug-associated mortality. 

I had only one substantive suggestion for the authors. I wonder if it is possible to use instrumental variables techniques here – this would require instruments, variables strongly associated with MD that are not associated with the confounding factors that the authors allude to. Is the case for instance that the local availability of narcotics (being close to international smuggling routes) affects MD. If so then this variable might be profitably used to implement an IV strategy. Certainly the sample size is large enough to do so.

RESPONSE: The reviewer suggests a promising line for future work. Other possible IV’s include geographic indicators of physician prescribing patterns and geographic patterns of industrial change. This additional research would have to be done at a sub-state geography. Unfortunately, it is beyond our scope at present.

REFERENCES

1. Ruhm CJ. Drug Mortality and Lost Life Years Among U.S. Midlife Adults, 1999-2015. Am J Prev Med. 2018 Jul;55(1): 11-8.

2. National Institute on Drug Abuse and Alcoholism. Alcohol Facts and Statistics. Available: https://www.niaaa.nih.gov/publications/brochures-and-fact-sheets/alcohol-facts-and-statistics.

3. Shiels MS, Berrington de Gonzalez A, Best AF, Chen Y, Chernyavskiy P, Hartge P, et al. Premature mortality from all causes and drug poisonings in the USA according to socioeconomic status and rurality: an analysis of death certificate data by county from 2000-15. Lancet Public Health. 2019 Feb;4(2): e97-e106.

---

## [Decision Letter · Decision Letter 1]

6 Dec 2019

Estimating the impact of drug use on US mortality, 1999-2016

PONE-D-19-25601R1

Dear Dr. Glei,

We are pleased to inform you that your manuscript has been judged scientifically suitable for publication and will be formally accepted for publication once it complies with all outstanding technical requirements.

With kind regards,

João Pedro Silva, PhD

Academic Editor

PLOS ONE

Additional Editor Comments (optional):

Reviewers' comments:

Reviewer's Responses to Questions

**Comments to the Author**

1. If the authors have adequately addressed your comments raised in a previous round of review and you feel that this manuscript is now acceptable for publication, you may indicate that here to bypass the “Comments to the Author” section, enter your conflict of interest statement in the “Confidential to Editor” section, and submit your "Accept" recommendation.

Reviewer #2: All comments have been addressed

2. Is the manuscript technically sound, and do the data support the conclusions?

Reviewer #2: Yes

3. Has the statistical analysis been performed appropriately and rigorously? 

Reviewer #2: Yes

4. Have the authors made all data underlying the findings in their manuscript fully available?

Reviewer #2: Yes

5. Is the manuscript presented in an intelligible fashion and written in standard English?

Reviewer #2: Yes

6. Review Comments to the Author

Reviewer #2: (No Response)

7. PLOS authors have the option to publish the peer review history of their article (what does this mean?). If published, this will include your full peer review and any attached files.

Reviewer #2: No

---

## [Editor Report · Acceptance letter]

12 Dec 2019

PONE-D-19-25601R1 

Estimating the impact of drug use on US mortality, 1999-2016 

Dear Dr. Glei:

I am pleased to inform you that your manuscript has been deemed suitable for publication in PLOS ONE. Congratulations! Your manuscript is now with our production department. 

With kind regards,

on behalf of

Dr. João Pedro Silva 

Academic Editor

PLOS ONE